# Virus Etiology, Diversity and Clinical Characteristics in South African Children Hospitalised with Gastroenteritis

**DOI:** 10.3390/v13020215

**Published:** 2021-01-30

**Authors:** Esmari Rossouw, Marieke Brauer, Pieter Meyer, Nicolette M. du Plessis, Theunis Avenant, Janet Mans

**Affiliations:** 1Department of Medical Virology, Faculty of Health Sciences, University of Pretoria, Pretoria 0001, South Africa; u12225925@tuks.co.za; 2Immunology Laboratory, Ampath, Pretoria 0001, South Africa; brauerm@ampath.co.za; 3Department of Immunology, Faculty of Health Sciences, University of Pretoria, Pretoria 0001, South Africa or pieter.meyer@nhls.ac.za; 4National Health Laboratory Service, Tshwane Academic Division, Pretoria 0001, South Africa; 5Department of Paediatrics, Kalafong Provincial Tertiary Hospital, Faculty of Health Sciences, University of Pretoria, Pretoria 0001, South Africa; nicolette.duplessis@up.ac.za (N.M.d.P.); theunis.avenant@up.ac.za (T.A.)

**Keywords:** paediatric gastroenteritis, rotavirus, norovirus, *FUT2* secretor status, HIV exposure

## Abstract

Background: Viral gastroenteritis remains a major cause of hospitalisation in young children. This study aimed to determine the distribution and diversity of enteric viruses in children ≤5 years, hospitalised with gastroenteritis at Kalafong Provincial Tertiary Hospital, Pretoria, South Africa, between July 2016 and December 2017. Methods: Stool specimens (*n* = 205) were screened for norovirus GI and GII, rotavirus, sapovirus, astrovirus and adenovirus by multiplex RT-PCR. HIV exposure and *FUT2* secretor status were evaluated. Secretor status was determined by *FUT2* genotyping. Results: At least one gastroenteritis virus was detected in 47% (96/205) of children. Rotavirus predominated (46/205), followed by norovirus (32/205), adenovirus (15/205), sapovirus (9/205) and astrovirus (3/205). Norovirus genotypes GI.3, GII.2, GII.3, GII.4, GII.7, GII.12, GII.21, and rotavirus strains G1P[8], G2P[4], G2P[6], G3P[4], G3P[8], G8P[4], G8P[6], G9P[6], G9P[8] and sapovirus genotypes GI.1, GI.2, GII.1, GII.4, GII.8 were detected; norovirus GII.4[P31] and rotavirus G3P[4] predominated. Asymptomatic norovirus infection (GI.3, GI.7, GII.4, GII.6, GII.13) was detected in 22% of 46 six-week follow up stools. HIV exposure (30%) was not associated with more frequent or severe viral gastroenteritis hospitalisations compared to unexposed children. Rotavirus preferentially infected secretor children (*p* = 0.143) and norovirus infected 78% secretors and 22% non-secretors. Conclusion: Rotavirus was still the leading cause of gastroenteritis hospitalisations, but norovirus caused more severe symptoms.

## 1. Introduction

Diarrhoeal disease, including viral gastroenteritis, is the second leading cause of infectious disease morbidity and one of the top 10 leading causes of mortality worldwide [1]. In the 2017 Global Burden of Disease study, diarrhoeal disease accounted for approximately 8% (or ~499,000) of global deaths, and mortality rates due to diarrhoeal disease in children under the age of 5 years were highest in sub-Saharan Africa and South-East Asia [2]. Viruses are amongst the most common causes of both endemic and epidemic gastroenteritis, with rotavirus, norovirus and adenovirus as the main causative agents, and sapovirus and astrovirus as contributing viruses [3]. The severity of rotavirus infection has decreased dramatically with the introduction of rotavirus vaccines, especially in high-income countries [4], resulting in lower rates of rotavirus hospitalisations (6.1% in USA [5]) and norovirus becoming the leading cause of paediatric gastroenteritis hospitalisations in some countries. The prevalence of rotavirus in hospitalisations in low- to middle-income countries such as South Africa (SA), however, is still estimated at 11−19% [6,7], with G3P[8] being the predominant strain detected [7]. The 2018 annual South African surveillance review [7] reported diarrhoeal disease data from four sentinel sites in three provinces. The surveillance detected rotavirus in 11%, norovirus in 21%, adenoviruses in 12%, sapoviruses in 6% and astroviruses in 3% of children hospitalised with diarrhoea at the sentinel sites [7].

The overall prevalence of noroviruses in children with gastroenteritis in low-income countries in Africa has been estimated to be 13.5% [8]. From 2009 to 2013 noroviruses were detected in 15% of children hospitalised with gastroenteritis in SA [9], with norovirus GII.4 being the predominant genotype and a high prevalence of recombinant norovirus strains [10]. Due to the large disease and economic burden of noroviruses, an effective vaccine is in demand for high-risk populations, such as young children, the elderly, and immunocompromised individuals [11]. Understanding the prevalence and diversity of asymptomatic norovirus infections is also important as asymptomatic individuals may facilitate the transmission of this virus [12].

Variation in pathogen-specific incidence between different geographical populations suggest that inherited host factors may differentially influence the susceptibility to different enteric infections. Host factors that may be important are the fucosyltransferase genes, *FUT2* and *FUT3*. Multiple studies have shown that secretor status, determined by the *FUT2* gene, can influence the risk of infections, specifically with some noro- and rotaviruses [13,14,15,16]. Several *FUT2* single nucleotide variations (SNV) have been identified. The nonsense variation G428A is representative of the dominating non-secretor allele (se428) in Europeans and Africans [17], but even though the frequencies of non-secretors in most populations are similar, the occurrence of Se enzyme deficiency, and the single-nucleotide variation associated is race specific [18,19]. The G428A SNV appears in approximately 20% of the European population and 30% of the African population [20,21]. Feeding habits (specifically breastfeeding) have also been proven to affect a child’s microbiome, which in turn may affect susceptibility to noro- and rotavirus infection [21,22,23]. Thus, there are multiple hypotheses of different pathways by which both the child’s *FUT2* and *FUT3* genes as well as those of the breastfeeding mother may alter the susceptibility to these enteric infections [24].

Approximately 7.7 million South Africans of all ages are infected with the human immunodeficiency virus (HIV), thus HIV status should be considered in any study in this population. The HIV transmission rate from mother to child in South Africa is approximately 3% [25], resulting in a large number of HIV exposed, uninfected (HEU) children. Various studies on the effects of HIV exposure have been performed in children. These effects included children having increased mortality rates, increased infection-related morbidity and impaired growth, when compared to HIV unexposed (HU) infants [26,27,28,29,30,31]. Multiple factors contribute to the differences between HEU and HU infants and these include maternal health, duration of maternal antiretroviral treatment, nutritional status and feeding practices, which are variable as breastfeeding recommendations for HIV infected (HI) mothers change over time [28,32,33].

Understanding the prevalence of and relationship between causative viruses of gastroenteritis and the factors that influence susceptibility and severity is essential to improve disease management and guide vaccine development [34]. More detailed characterisation of the interactions between viral pathogens and histo-blood group antigens (HBGAs) in different populations could improve the success of regionalised and targeted interventions [24]. This study aimed to determine the distribution and diversity of gastroenteritis viruses in South African children ≤5 years of age, and to investigate asymptomatic norovirus infections as well as the effect of HIV exposure and secretor status on viral gastroenteritis infections.

## 2. Materials and Methods

### 2.1. Ethics Approval

This study was approved by the Faculty of Health Sciences Research Ethics Committee of the University of Pretoria—Ref: 362/2015, 90/2017, 182/2018 (renewed on 10/04/2019).

### 2.2. Study Population

From July 2016 to December 2017, 221 children below the age of 5 years hospitalised with diarrhoea, at Kalafong Provincial Tertiary Hospital (KPTH), Pretoria, South Africa, were enrolled in this study. Informed consent was obtained from each parent/caregiver and demographic and clinical information as well as HIV status information, was collected with a questionnaire. Stool and whole blood specimens were collected from each participant. Parents/caregivers were requested to submit a follow up stool specimen 6 weeks after enrolment. Participants were divided into three categories, based on their HIV status, namely HI, HEU and HU. Severity of disease was categorised into either mild, moderate or severe categories using the Vesikari clinical severity scoring system [35,36,37].

### 2.3. Stool Specimen Preparation and Nucleic Acid Extraction

Stool specimens were collected from diapers and stored at −20 °C and 10% suspensions were prepared using phosphate buffered saline (PBS) (Thermo Scientific, Waltham, MA, USA). Automated total nucleic acid extraction was performed with 200 µL of 10% stool suspension supernatant using the automated NucliSENS^®^ EasyMAG^®^ Instrument (BioMérieux, Marcy-l’Étoile, France). The nucleic acids were eluted in 50 µL and stored at −80 °C. An internal control included in the virus screening kit was added to each specimen during each extraction experiment according to the manufacturer’s instructions.

### 2.4. Enteric Virus Detection

A multiplex reverse transcriptase real-time polymerase chain reaction (RT-PCR, Allplex^TM^ Gastrointestinal Virus Panel, Seegene Inc., Seoul, South Korea), was used to screen each specimen for six gastroenteritis viruses (norovirus GI & GII, sapovirus, rotavirus, adenovirus-F and astrovirus), on the Bio-Rad CFX platform (Bio-Rad Laboratories, Hercules, CA, USA). The one-step RT-PCR (20 µL) was performed according to the kit’s instructions, with the modification that the nucleic acids were incubated at 95 °C for 2 min (min) and then on ice for 2 min to ensure the separation of rotavirus’s double-stranded RNA before RT-PCR. Seegene Viewer analysis software (Seegene Inc., Seoul, South Korea) was used to analyse the results. Nucleic acids were extracted from all follow up specimens as described above and these were screened for norovirus GI and GII with a real-time RT-PCR using the Qiagen QuantiFast Pathogen RT-PCR and IC kit (Qiagen Inc., Hilden, Germany), according to manufacturer’s instructions, with specific primers and probes for norovirus GI and GII (Appendix A) [38,39,40,41,42].

### 2.5. Genotyping of Enteric Viruses

#### 2.5.1. Complementary DNA Synthesis

Complementary DNA (cDNA) was prepared for norovirus, sapovirus and rotavirus with ProtoScript II Reverse Transcriptase (New England Biolabs, Ipswitch, MA, USA) and random hexamers (Roche Diagnostics Corp., Mannheim, Germany) according to the manufacturer’s instructions using 10 uL of RNA.

#### 2.5.2. Calicivirus Genotyping

Norovirus strains were characterised by amplification and nucleotide sequencing of the partial RNA-dependent RNA polymerase (RdRp) and capsid genes. As a first step a ~1100 bp region spanning the 3′-end of the RdRp and the 5′-end of the major capsid gene (AC region) was targeted for amplification. If this region did not amplify, a smaller region spanning the ~560 bp BC region, was amplified (Appendix A). In some cases, the BC region also failed to amplify after which region C (~320 bp) was amplified. Sapovirus was genotyped based on characterisation of the partial capsid gene (Appendix A).

The various virus genes were amplified from 5–10 µL cDNA with EmeraldAmp^®^ MAX HS PCR Master Mix (Takara Bio Inc., Shiga, Japan) using the primers listed in Appendix A and conditions listed in Appendix A. PCR products were purified with the Zymo Research Clean and Concentrator−25 kit (Zymo Research, Irvine, CA, USA) and sequenced with the PCR primers (Appendix A) with the ABI PRISM BigDye^®^ Terminator v. 3.1 Cycle sequencing kit (Applied Biosystems, Foster City, CA, USA).

#### 2.5.3. Rotavirus Genotyping

The partial VP4 and VP7 gene fragments of rotaviruses were amplified with nested RT-PCR using the EmeraldAmp^®^ MAX HS PCR Master Mix (Takara Bio Inc., Shiga, Japan) according to manufacturer’s instructions and primers listed in Appendix A [43]. RT-PCR products (Appendix A) were purified and sequenced as described above.

### 2.6. Phylogenetic Analyses

Nucleotide sequences were analysed using the Sequencher DNA Sequence Analysis Software (Gene Codes Corporation, Ann Arbor, MI, USA), BioEdit Sequence Alignment Editor [44] and BLAST-n [45]. Norovirus, rotavirus and sapovirus sequences were aligned with reference strains using MAFFT version 7 (http://mafft.cbrc.jp/alignment/server/index.html) [46] and maximum likelihood phylogenetic analysis was performed with IQ-Tree version 1.6.12 [47], with the best fit model automatically selected with ModelFinder and the tree topology validated by 1000 ultrafast bootstrap replicates [48]. Genotypes were assigned based on clustering with reference strains in the phylogenetic tree with >70% bootstrap support. The RdRp and capsid genotypes for norovirus were determined using the Norovirus Genotyping Tool (https://www.rivm.nl/mpf/typingtool/norovirus/) [49]. The nucleotide sequences determined in this study were submitted to GenBank under the following accession numbers: norovirus RdRp (GI—MW386172; GII—MW389666-MW389684), norovirus capsid (GI—MW385950-MW385953; GII MW381792-MW381823), rotavirus VP4 (MW392000-MW392041), rotavirus VP7 (MW392042-MW392081), sapovirus (MW391992-MW391999) and *FUT2* (MW491814-MW491833).

### 2.7. Total Genomic DNA Extraction from Whole Blood Samples

Manual genomic DNA extraction was performed on 190 (200 μL) of the whole blood specimens using a QIAamp DNA Blood Mini kit (Qiagen Inc., Hilden, Germany), to a final elution volume of 200 µL, as per manufacturer’s instructions. After extraction DNA was stored at −20 °C until use.

In the case of no blood specimen being available (15/205), total nucleic acids were extracted from stool samples using the NucliSENS^®^ EasyMAG^®^ automated nucleic acid extraction instrument (BioMérieux, Marcy-l’Étoile, France) and stored at −80 °C until use.

### 2.8. Nonsense Variation Detection Using Real-Time PCR and SNP Assay

The TaqMan SNP assay kit (SNP ID: rs601338, Applied Biosystems, Foster City, CA, USA) was used to detect the G428A nonsense variation as per manufacturer’s protocol. The master mix was initially prepared to a final volume of 10 µL, as stated in the protocol, but after encountering discrepant results, the master mix was doubled per sample (final volume = 20 µL).

### 2.9. FUT2 Genotyping of Non-Secretor Samples

Due to misclassification of the non-secretor genotype in some samples by the TaqMan assay [50], non-secretor genotype results were confirmed by nucleotide sequencing of the *FUT2* gene. The 1263 bp exon 2 of the *FUT2* gene was amplified by PCR using Q5^®^ Hot Start High-Fidelity DNA Polymerase (New England Biolabs, Ipswitch, MA, USA) and published primers (Appendix A) [17]. Nucleotide sequences were analysed using the Sequencher DNA Sequence Analysis Software (Gene Codes Corporation, Ann Arbor, MI, USA). Sequences containing the G allele at the 428 bp position were classified as homozygous secretors (SeSe), sequences containing an A at this position were homozygous non-secretors (sese), and sequences containing both A and G at this position were determined to be heterozygous secretors (Sese).

### 2.10. Statistical Analyses

Descriptive and inferential statistics techniques were used in the analyses. Tests for association of contingency tables was performed using two-tailed Chi^2^ test and Fisher’s exact test for smaller numbers. Dunn’s test of multiple comparisons using rank sums were used with Bonferroni for multi-comparison adjustment for water source and HIV exposure. Multinomial regression was applied to detect possible associations with viruses and symptoms. Statistical significance was determined by *p*-value < 0.05. The analyses were done using STATA 16.1 (StataCorp LLC, College Station, TX, USA).

## 3. Results

### 3.1. Study Population

A total of 221 children ≤5 years, hospitalised with gastroenteritis from July 2016 to December 2017 were enrolled. Stool specimens could be collected for 205 children and they were included in the study. The median age of the study population was 10 months (13 days minimum, 60 months maximum), with 82% of specimens (169/205) collected from children ≤2 years of age, and the largest contribution from children between 7 and 12 months. The male:female ratio was 1.5:1. The demographic and clinical characteristics of the cohort are summarised in Table 1. Most of the children were HIV unexposed (*n* = 134; 65%), with HIV exposed children accounting for 30% (*n* = 61) of the study population and HIV positive children for 5% (*n* = 10). *FUT2* genotyping determined 81% secretors versus 19% non-secretors in this population. The children comprised of 57 homozygous secretors, 110 heterozygous secretors and 38 homozygous non-secretors. The majority of children (71%) exhibited diarrhoeal symptoms for between 1 to 4 days, with 43% of children experiencing >6 loose/watery stools in 24 h. In total, 58% of children vomited and 37% had a fever in the 48 h prior to hospitalisation.

HIV exposure did not affect the frequency of gastroenteritis virus infections in this cohort (*p* > 0.5). A general trend in increasing ratios of co-infections was observed from HU to HEU to HI. Due to the small HI sample size, this was not statistically significant. The severity of disease was also evaluated against HIV status. The difference in severity of disease for HI, HEU and HU patients was not statistically significant (*p* = 0.072). There was no association between viral gastroenteritis and type of water source (indoor tap compared to outdoor sources, *p* = 0.1993) and type of sanitation (flush toilet versus other types, *p* = 0.0987).

### 3.2. Virus Detection

A total of 47% (96/205) of children were infected with at least one gastroenteritis virus. Virus-infected children were significantly younger (median age 9 months) compared to virus-negative children (median age 14 months, *p* = 0.0002). Rotavirus was the predominant virus detected (*n* = 46; 22%) despite 37/46 children being recorded as fully vaccinated against the virus, this was followed by norovirus GII (*n* = 29; 14%) and then adenovirus (*n* = 15; 7%), sapovirus (*n* = 9; 4%), norovirus GI and astrovirus (*n* = 3; 1.5% each) (Figure 1a). Additionally, 9% virus co-infections (9/96) were observed. All viruses were detected in children under the age of 2 years, except one adenovirus, two norovirus, and four rotavirus infections that were detected in children aged between 24 months and 5 years (Figure 1b).

Bacterial and parasite co-infection data were obtained from the National Health Laboratory Service, where microscopy, culture and sensitivity (MCS) testing was performed on specimens from a total of 183/205 patients. A total of 22 patients’ MCS data were unavailable. A total of five parasite (1.6%) and 10 bacterial infections (5.5%) were observed, with five viral and microbial co-infections.

Follow up stool specimens were obtained for 46/205 patients. In total, 22% (10/46) of the specimens tested positive for norovirus GI (2) or GII (8), with all these participants being asymptomatic at the time the follow up specimen was collected. Disease severity scores and detected virus were compared for all the children, to determine if specific virus infections presented with more severe symptoms. Figure 2 indicates that norovirus GII infection most often coincided with severe illness. Viral co-infections also presented with more severe symptoms (Figure 2). The child with a single astrovirus infection also presented with severe symptoms, although this should be interpreted with caution, due to the small sample size.

### 3.3. Virus Genotyping

Norovirus was detected in 15.6% (32/205) of specimens. Genogroups GI and GII represented 9.4% (3/32) and 90.6% (29/32) of norovirus cases, respectively, with no GI/GII co-infections detected. In total, 28/32 (2 GI, 26 GII) norovirus strains could be genotyped, representing seven genotypes, including emerging strains such as GII.2 and GII.21, with GII.4 being predominant (Figure 3a). Of the 10 asymptomatically infected patients detected during follow up, two were infected with norovirus at enrolment. A total of 9/10 of the strains could be genotyped, representing five genotypes, with GII.4 and GII.13 being the most prevalent (Figure 3b). A total of 44/46 rotavirus strains were genotyped, representing three P types and five G types, with a total of nine different combinations, of which G3P[4] predominated (Figure 3c). The majority of sapovirus strains (8/9) could be amplified and genotyped, resulting in five genotypes (including a GII.8, which represents the first report of this genotype in South Africa) being identified among the eight specimens with GI.2 being the most predominant (Figure 3d).

### 3.4. Phylogenetic Analysis

Phylogenetic analyses of the partial capsid and RdRp regions of the norovirus study strains and the most closely related strains identified in GenBank are shown in Figure 4a,b. A total of seven capsid types and seven RdRp types were determined, with GII.4[P31] (15/27, 56%) observed most often. The other strains were detected at between 4% and 7%. The partial GII.4 capsid sequences formed two clusters correlating with their respective RdRp types [P31] or [P16]. The majority of GII.4[P31] strains (11/15) were very closely related. Two GenBank strains that were detected in 2016 in Australia and China represent a large number of strains with 99–100% identity over the analysed region identified by BLAST analysis. The SA strains were closely related to noroviruses detected in the Americas, Europe, Asia and Australia.

Phylogenetic analysis of the noroviruses detected at 6-week follow up, indicated that one child was infected with the same strain (NS0173—GII.4) at both time points. The other child, for which both the original and follow up strains could be genotyped, was initially infected with GII.4 followed by a GI.7 (NS0017) infection (Figure 5).

The rotavirus strains detected in this study were closely related to each other within each G (VP7) and P (VP4) type. Phylogenetic analysis of the G type sequences indicated two clusters in each G type, with several identical sequences in each cluster (Figure 6a). The P type analysis showed a similar high level of identity between study strains (Figure 6b). The P[6] type sequences of three rotavirus strains (NS0015, NS0035 and NS0199) that infected non-secretor children were identical (over 555 bp) to the nine other P[6] strains that were all identified in secretor children.

Phylogenetic analysis of sapovirus strains detected in this study showed that closely related strains infected both secretor and non-secretor children (Figure 7). The South African sapovirus strains were related to strains circulating in China (GII.2), Peru (GI.1, GII.4 and GII.8), Japan (GII.1) and Spain (GII.4).

### 3.5. Secretor Status and Virus Genotypes

The secretor status of 24/38 non-secretor children (13/17 virus-infected non-secretor children) could be confirmed with nucleotide sequencing of the *FUT2* gene. Amplification of the *FUT2* gene failed from the stored blood samples of the remaining 14 children. The virus genotypes identified in the children were correlated with their secretor status (*FUT2* genotype). In total, 78% (25/32) of the children infected with either norovirus GI or GII were secretors, whereas 22% (7/32) were non-secretors (Figure 4). In the non-secretor population, five genotypes were identified (GI.3, GII.2[P2], GII.3[PNA], GII.4[P31], GII.21[P21]), while three specimen’s strains could not be determined due to low viral load. Six genotypes were identified in the secretor children (GI.3[P3], GII.2[P2], GII.4[P16], GII.4[P31], GII.7, GII.12[P33]), and the GII.4 genotype represented 75% (18/24 typed) of the norovirus strains, whereas in non-secretors it constituted 25% (1/4 typed) of strains.

When comparing rotavirus positive patients, 91% (42/46) of the total population were secretors, whereas 9% (4/46) were non-secretors, indicating a significant preference of rotavirus for secretors (*p* = 0.0143). Three genotypes were identified in the non-secretor population, with three of the non-secretor children infected with rotavirus P[6] (1x G2P[6], 2x G8P[6]). For the secretor-positive children infected with rotavirus, nine strains were determined, with G3P[4] (17/42) predominating. When the other virus-positive (sapovirus and adenovirus) patients’ *FUT2* genotyping analysis was performed, these ratios were closer to 1:1, indicating no specific preference of these viruses for secretors or non-secretors. The astrovirus patient subset was too small to make any significant conclusions (Figure 8).

## 4. Discussion

Viral gastroenteritis is still a major cause of illness, in the face of improvements to provision of safe water and sanitation, the rotavirus vaccine and reductions in hospitalisations and mortality [51,52,53]. In high-income countries, the rotavirus vaccine has led to a rapid decrease in hospitalisations due to rotavirus, with norovirus now the major cause of viral gastroenteritis hospitalisations in children under the age of 5 [4,54]. However, in low- and middle-income countries such as South Africa, despite routine rotavirus vaccination, rotavirus is still the leading cause of viral gastroenteritis hospitalisations [55], as was illustrated by this study.

In-depth understanding of the epidemiology of gastroenteritis viruses is crucial to help with the fight against severe viral infections. This 18-month study focused on severe gastroenteritis at a single hospital KPTH, in a relatively urban setting in Pretoria, Gauteng, South Africa’s most populous province. Of the 205 children admitted in the study, the majority (119/205) were under the age of 18 months, underscoring that gastroenteritis is more severe in younger children [56,57]. One aspect of the study was to evaluate whether HIV exposure has an effect on the frequency or severity of viral gastroenteritis infections. Thirty percent of the study population was HIV exposed, but the virus infection rate and level of severity were no different in the HEU children compared to HU children (*p* = 0.072). This correlates with data from a recent study on the burden of diarrhoeal disease in children at Chris Hani Baragwanath Hospital (CHBH) in the Johannesburg region [58]. Between 2015 and 2016, Makgatho and colleagues reported HIV exposure in 36.5% of children younger than 6 months, with lower median weight-for-age and height-for-age Z-scores in HEU compared with HU infants, but with no increase in hospital stay or higher prevalence of coexisting lower respiratory tract infections [58]. HIV-infected children had increased odds of death (aOR = 9.1) whereas HEU infants did not. Taken together the data indicate that HIV exposure does not negatively impact on hospitalised cases of viral gastroenteritis in South African children.

Rotavirus (22%), norovirus (15.6%), adenovirus-F (7%) and sapovirus (4%) were the most common viral pathogens detected at KPTH. Astrovirus was mostly observed along with rotavirus infection (2/3), consistent with co-infection rates of 82.5% in South African children with astrovirus infections [59].

A total of 46 children (22%) were infected with rotavirus (40 single infections (20%), 6 co-infections). This is a high prevalence when compared to high income countries, with countries such as the United States reporting a median detection rate of 1% (ranging between 0–3.4%) in 2020 [60]. In lower income countries such as SA, a higher prevalence is more common [61,62], with a recent study indicating rotavirus prevalence in hospitalised children at 11% [7]. Interestingly severe gastroenteritis was observed in 66% of norovirus cases (21/32) whereas 62.5% (25/40) of the single rotavirus infections were in the mild to moderate category. Of the 46 rotavirus-positive children, 37 had been fully vaccinated against rotavirus, the majority with Rotarix (G1P[8]). This indicates that the vaccine could have reduced the severity of infection, but did not prevent hospitalisation. A similar observation was made in Burkina Faso, after rotavirus vaccine introduction, between 2015 and 2016, where norovirus caused more severe gastroenteritis than rotavirus in hospitalised children. In contrast to the KPTH study, norovirus was detected at a higher frequency (20%) than rotavirus (14%) in Burkina Faso [63].

Data on asymptomatic norovirus infection is limited in SA. The current study’s results are comparable to a previous study in SA, which detected 36% asymptomatic norovirus infections [64], as well as other studies in low-income settings, such as the Garcia et al. study that found almost 30% norovirus positive specimens from asymptomatic patients in Mexico [65]. The higher frequency of asymptomatic infections in the Kabue et al. study is likely due to the more rural setting, which has been shown to be associated with a higher prevalence of norovirus transmission [66,67].

Nine viral co-infections were observed (4%), with an additional three bacterial/parasite co-infections. These ratios are lower than those observed in other studies, especially in middle- to low-income countries, with co-infections ranging from 1.1% to 76.7% with a median of 26.7% [68]. The time between stool specimen collection and MCS testing was variable throughout the study and thus the bacteria/parasite co-infection rate could have been underestimated.

Overall, the virus genotyping success rate was high in this study ranging from 87.5% for noroviruses to 96% for rotaviruses. Various rotavirus G and P type combinations were observed with the most prevalent combination being G3P[4] (17/46; 37%) followed by G3P[8] (13%) and G2[P6] (13%). Several studies have monitored rotavirus strain circulation before and after vaccine introduction. A five-year study (2010−2015) in East and Southern Africa found no difference in circulating strains before and after introduction, with fluctuations of strains observed over time [69]. Page and colleagues found a significant decrease in G1 strains post vaccine introduction in SA (2009–2014), with a corresponding increase in non G1P[8] strains [70]. A recent study from Mozambique reported the emergence of G3P[4] and G3P[8] in the post vaccine era, with G3P[4] increasing from 2.6% prevalence in 2016 to 38.7% in 2018 [71], correlating with the predominance of G3P[4] observed in 2016/2017 at KPTH. More recent studies have now included whole-genome sequencing of rotaviruses [72,73], to give a better overall perspective on the specific strains detected, their origin and reassortment, this should be considered for future studies, to improve our understanding of rotavirus epidemiology.

The majority of the norovirus GII strains were determined to be GII.4s (19/29), which was to be expected, as this is the most prevalent and virulent norovirus genotype [74,75,76]. The Sydney 2012 variant, with two different RdRps, GII.P31 (previously GII.Pe) and GII.P16 have circulated during the study [77]. Other genotypes identified during this study include emerging types, such as GII.2 (*n* = 2), which has been detected frequently in wastewater, but not yet as commonly in clinical specimens in South Africa [78]. In terms of the follow up specimens, five different genotypes were observed, with GII.4 and GII.13 being the most prevalent. When comparing the norovirus genotypes detected in the children at initial hospitalisation (GI.3, GII.2, GII.3, GII.4, GII.7, GII.12, GII.21) with the genotypes in asymptomatic infections (GI.3, GI.7, GII.4, GII.6, GII.13) a different distribution was observed, with only GI.3 and GII.4 detected in both groups. It is possible that the children with asymptomatic GII.4 infection at follow up may have had earlier symptomatic GII.4 infections. The median age of children asymptomatically infected with norovirus was 11.5 months compared to 10 months for children with symptomatic infection.

Sapovirus infections were detected in 4% of children at KPTH and the most prevalent genotype identified was GI.2 (3/9). From 2009−2013, a sentinel hospital surveillance study in SA detected sapovirus in 7.7% (range 6.3–8.7%) of hospitalised children [79] and reported GIV strains as predominant [80]. The observed differences are likely due to the relatively small cohort in the current study. A GII.8 strain was described for the first time in South Africa in this study and it is most closely related to a GII from South Africa identified in 2014. (GenBank accession number: KP196511.1).

The rota-, noro- and sapoviruses that are circulating in South African children in Gauteng show high levels of similarity to strains reported across the world. Rotavirus strains were closely related to genotypes reported in various Southern African (SA, Malawi, Mozambique, Zambia, Zimbabwe), West African (Ghana) and East African (Ethiopia, Kenia, Uganda) countries between 2008 and 2016. In part this reflects the success of the African Rotavirus Network to generate rotavirus genotype data from Africa [81]. Various closely related strains were also reported in Europe and Asia. The SA norovirus strains were closely related to strains reported in the Americas, Europe, Asia and Australia between 2012–2020. Apart from closely related strains identified in SA, the only other African country reporting similar strains was Cameroon. This could be due to limited norovirus surveillance in most African countries.

The impact of secretor status on susceptibility to rotavirus and norovirus infections has been increasingly recognised in recent years [82,83]. In this study, rotavirus infection was observed at a ratio of 10:1 for secretors and non-secretors, indicating that rotavirus preferentially infects secretors (*p* = 0.0143). The majority of norovirus infections (25/32) were detected in secretor children (78%) with 22% of infections in non-secretors. The infection rate for both rota- and norovirus in non-secretor South African children is higher than in Brazilian children where only 3% of rotavirus infections and 1.5% of norovirus infections were detected in non-secretors [84]. In the Brazilian study, norovirus GII.3 infected a non-secretor and GII.4 exclusively infected secretors. In SA, non-secretors were infected with GI.3, GII.2, GII.3, GII.4 and GII.21 and two non-secretor infections could not be genotyped. Whether this difference reflects population HBGA differences or norovirus circulation pattern differences is not clear. Secretor status did not impact on adeno-, astro- or sapovirus infections, as has been observed in previous studies [24,85,86,87]. The noro- and rotavirus genotypes observed in the study were compared between secretors and non-secretors. GII.4 infections were detected almost exclusively in secretors (18/19), with only one non-secretor, NS0220, infected with GII.4 Sydney 2012 P[31]. This child had moderate symptoms and the norovirus viral load was low (Ct = 38). GII.4 is known to preferentially infect secretors, but exceptions have been observed [20,88,89,90,91]. The reasons for these exceptions are not yet clear, but could include microbiota diversity, such as HBGA-expressing bacteria [92], environment or feeding habits [24], differences between GII.4 variants, general health of the child, weak-secretor phenotype, or unidentified host factors [83], indicating that more studies are needed to gain a better understanding of these mechanisms. Rotavirus P[4] and P[8] genotypes bind to secretor HBGAs, such as Le(b) and H-type 1, while the P[6] genotype binds to type I HBGA precursors and H-type I and are more likely to infect Lewis negative individuals [82,93,94]. In this study, P[4] and P[8] genotypes mainly infected secretors, with one P[8] infection in a non-secretor (NS0207). P[6] infections represented 26% of rotavirus infections, the median age of P[6] infected children was 7 months and three of these infections were in non-secretors. These data correlate with higher levels of P[6] infections reported in Africa [95] and the tendency of P[6] strains to infect younger infants [96].

Limitations of this study include a relatively small sample size, specifically regarding HIV infection, which did not allow any meaningful analysis of the impact of HIV infection on viral gastroenteritis. In addition, the percentage of follow up stool specimens that were received for analysis of asymptomatic norovirus infections was less than 25%, which may skew the results and conclusions on asymptomatic infections. Funding and time constraints precluded genotyping of adeno- and astroviruses. Finally, it would have been helpful to determine the *FUT3* genotype of the cohort in addition to *FUT2*, to better comprehend the relationship between rotavirus infections and HBGAs.

In conclusion, rotavirus is still the leading cause of viral gastroenteritis hospitalisations in Gauteng, SA. However, norovirus seems to cause the majority of severe infections in young children post rotavirus vaccine introduction. HIV exposure does not predispose infants to more frequent and severe gastroenteritis episodes. The secretor to non-secretor ratio of 4:1 is comparable to other population groups outside of Africa and impacts on pathogen susceptibility and the circulation of specific noro- and rotavirus strains. Continued surveillance of gastroenteritis viruses is needed to monitor the situation regarding rota- and norovirus predominance in SA and future studies should consider the HBGA genotype and phenotype together with virus genotype to clarify the complex observations regarding virus susceptibility to ultimately improve vaccine design.

## Figures and Tables

**Figure 1 viruses-13-00215-f001:**
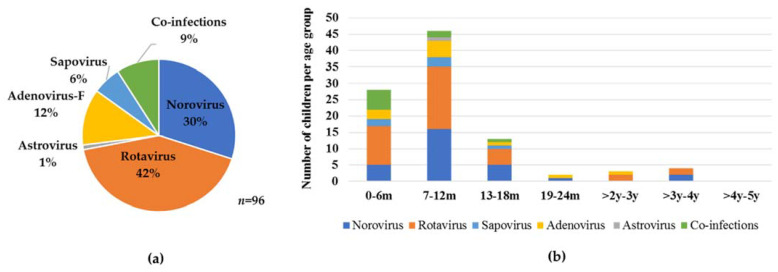
Virus distribution in (**a**) 96 virus infected patients and (**b**) within different age groups of the 205 children ≤5 years hospitalised with gastroenteritis at Kalafong Provincial Tertiary Hospital (KPTH) from July 2016 to December 2017.

**Figure 2 viruses-13-00215-f002:**
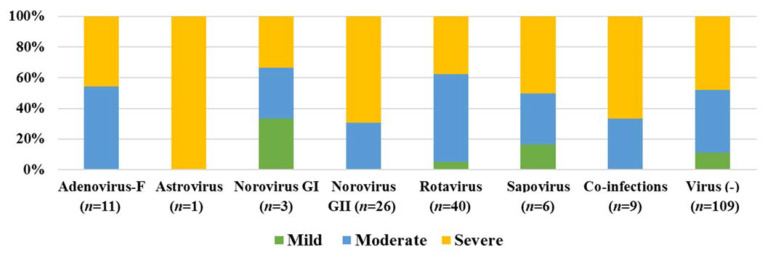
Disease severity observed during gastroenteritis virus infections (96) and virus unrelated gastroenteritis episodes (109) in children ≤5 years, hospitalised with gastroenteritis at KPTH from July 2016 to December 2017.

**Figure 3 viruses-13-00215-f003:**
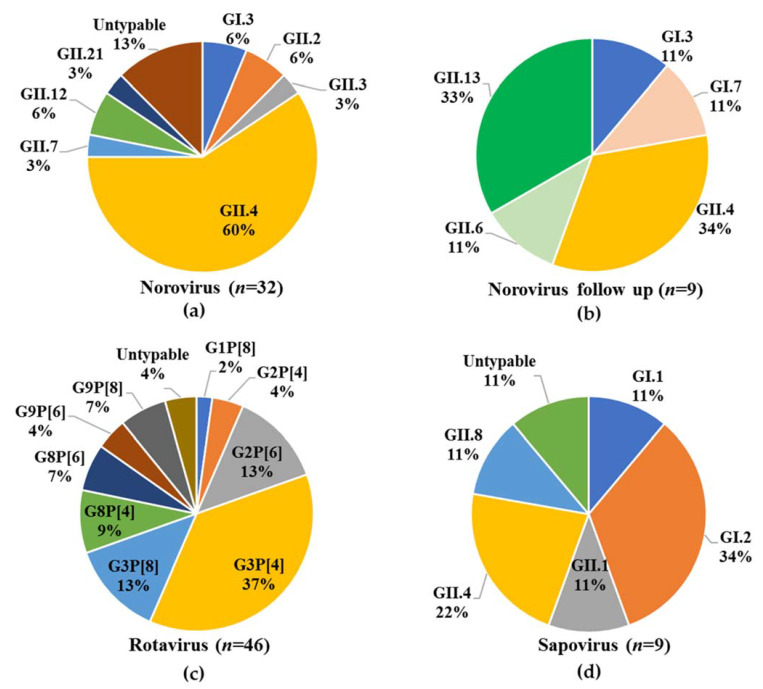
Virus diversity in children (≤5 years) hospitalised with gastroenteritis at KPTH from July 2016 to December 2017. (**a**) Norovirus genotype distribution in 32/205 children, (**b**) Norovirus genotype distribution in asymptomatic children (*n* = 9) 6 weeks after their initial hospitalisation with gastroenteritis, (**c**) Rotavirus genotype distribution in 46/205 children, (**d**) Sapovirus genotype distribution in 9/205 children.

**Figure 4 viruses-13-00215-f004:**
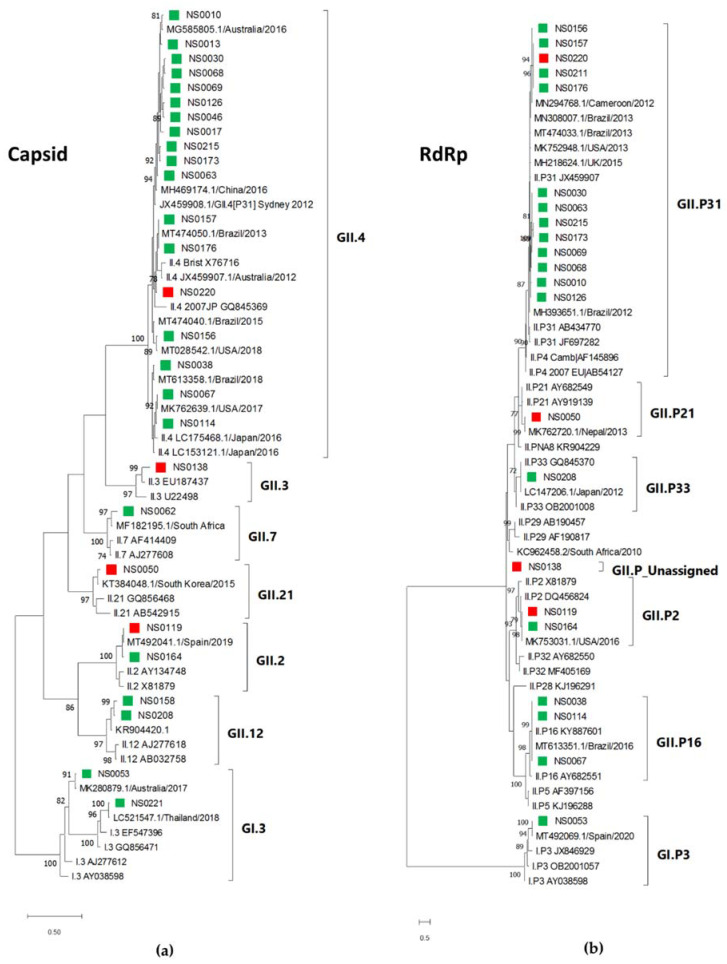
Maximum likelihood phylogenetic tree of (**a**) the partial capsid gene (GI—222 bp, GII—210 bp) and (**b**) the partial RdRp gene (198 bp) of the norovirus strains detected in children hospitalised with gastroenteritis at KPTH between July 2016 and December 2017. Closely related strains from GenBank are indicated by accession numbers. Bootstrap support of >70% is shown. The scale bar represents nucleotide substitutions per site. Secretor status of infected child is indicated by green (secretor) and red (non-secretor) blocks.

**Figure 5 viruses-13-00215-f005:**
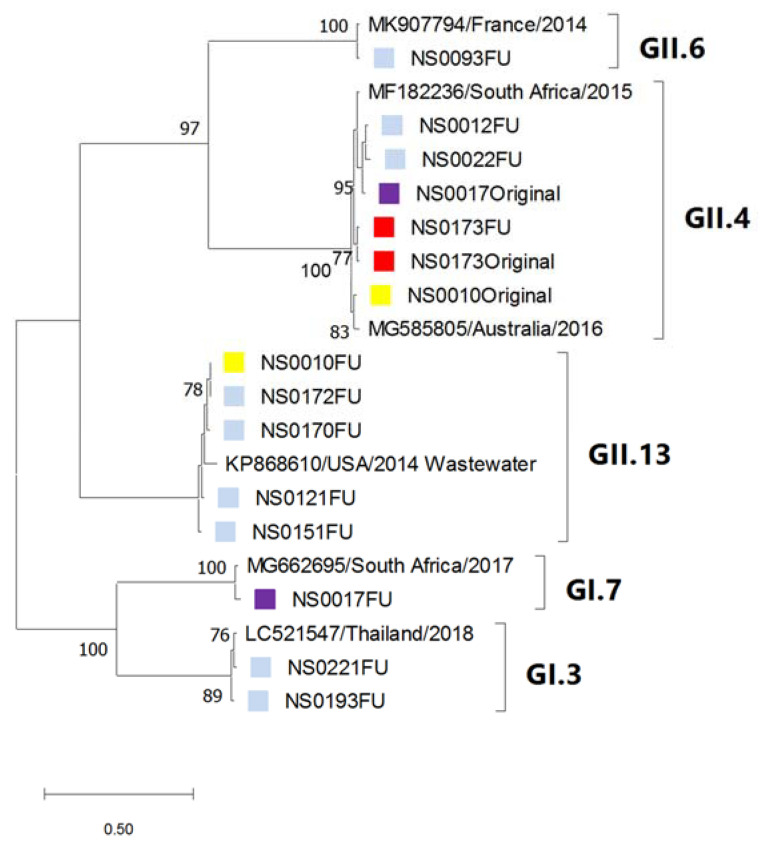
Comparison of norovirus strains detected in symptomatic and asymptomatic children. Maximum likelihood phylogenetic tree of the partial capsid gene (GI—288 bp; GII—276 bp) of the norovirus strains detected in children six weeks after initial hospitalisation with gastroenteritis at KPTH between July 2016 and December 2017 (blue blocks). The norovirus strains that were detected in two children at initial hospitalisation and at 6-week follow up are shown in matched colours (NS0017—yellow; NS0173—red). Closely related strains from GenBank are indicated by accession numbers. Bootstrap support of >70% is shown. The scale bar represents 0.5 nucleotide substitutions per site over the indicated region.

**Figure 6 viruses-13-00215-f006:**
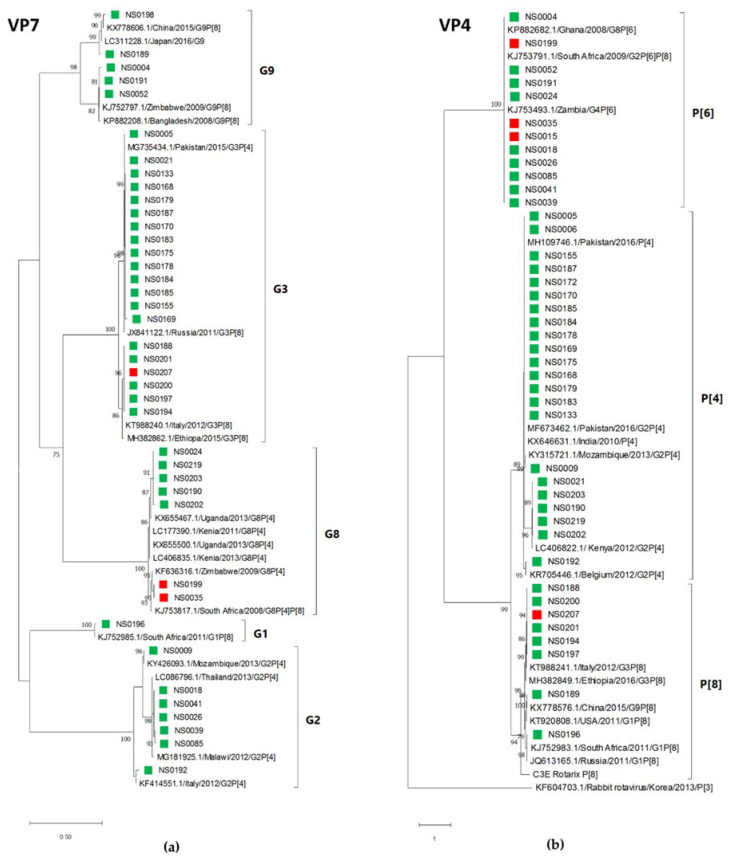
Rotavirus G and P types identified by maximum likelihood phylogenetic analysis of (**a**) the VP7 gene (585 bp) and (**b**) the VP4 gene (555 bp) of the rotavirus strains detected in children hospitalised with gastroenteritis at KPTH between July 2016 and December 2017. Closely related strains from GenBank are indicated by accession numbers. Bootstrap support of >70% is shown. The scale bar represents nucleotide substitutions per site. Secretor status of infected child is indicated by green (secretor) and red (non-secretor) blocks.

**Figure 7 viruses-13-00215-f007:**
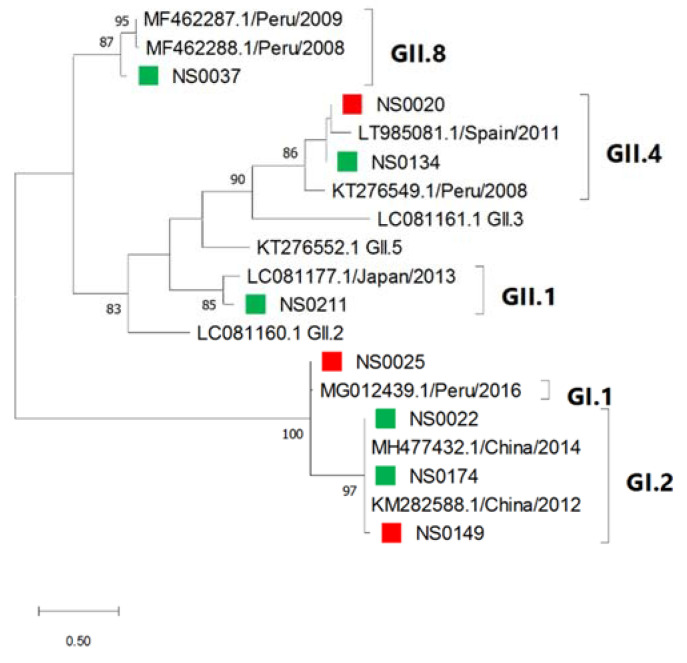
Sapovirus genotypes identified by maximum likelihood phylogenetic analysis of the partial capsid gene (207 bp) of sapovirus strains detected in children hospitalised with gastroenteritis at KPTH between July 2016 and December 2017. Closely related strains from GenBank were included. Bootstrap support of >70% is shown. The scale bar represents 0.5 nucleotide substitutions per site. Secretor status of infected child is indicated by green (secretor) and red (non-secretor) blocks.

**Figure 8 viruses-13-00215-f008:**
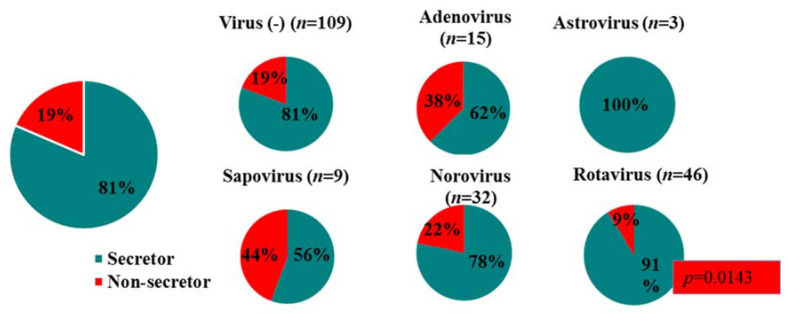
Secretor/non-secretor ratios for children infected with the different gastroenteritis viruses.

**Table 1 viruses-13-00215-t001:** Demographic and clinical characteristics of children with acute gastroenteritis (*n* = 205).

Characteristic	Cohort (*n* = 205)	Norovirus (+) Cohort (*n* = 32)	Rotavirus (+) Cohort (*n* = 46)	Virus (−) Cohort (*n* = 109)	Features Tested for Association	*p*-Value
**Demographic Characteristics**						
Median age (in months) at entry (min-max)	10 (0.5−64)	10 (0.5−46)	8.5 (0.5−48)	14 (1−64)	^a^ Age/Virus infection	0.0002 chi^2^ = 13.845
0–<12, n (%)	115 (56)	21 (66)	32 (70)	48 (44)		
12–<24, n (%)	51 (25)	9 (28)	10 (22)	29 (27)		
24–<48, n (%)	28 (14)	2 (6)	3 (7)	22 (20)		
48–60, n (%)	12 (16)	0	1 (1)	10 (9)		
Gender, n (%)						
Male	122 (60)	17 (53)	26 (57)	64 (59)		
Female	83 (40)	15 (47)	20 (43)	45 (41)		
Race, n (%)						
White	1 (0.5)	1 (3)	1 (2)	0		
Black	201 (98)	31 (97)	45 (98)	106 (97)		
Coloured	2 (1)	0	0	2 (2)		
Asian	1 (0.5)	0	0	1 (1)		
**Environmental Features**						
Water source					^a^ Water/	
Indoor tap, n (%)	79 (39)	15 (47)	24 (52)	36 (33)	Virus	0.1993
Other, n (%)	126 (61)	17 (53)	22 (48)	71 (65)	infection	chi^2^ = 1.647
Sanitation type					^a^ Sanitation/	
Flush toilet, n (%)	126 (61)	16 (50)	34 (74)	65 (60)	Virus	0.09873
Other, n (%)	79 (39)	16 (50)	12 (26)	44 (40)	infection	chi^2^ = 0.026
**Clinical characteristics**						
No. of days with diarrhoea, median (IQR)	3 (3)	3 (2)	3 (3)	3 (3)		
1–4 days, n (%)	145 (71)	25 (78)	35 (75)	36 (33)		
5 days, n (%)	21 (10)	2 (6)	3 (7)	25 (23)		
≥6 days, n (%)	39 (19)	5 (16)	8 (17)	48 (44)		
Maximum no. of loose/watery stools in 24 h period, median (IQR)	5 (3)	5 (3.75)	5 (3)	5 (2.5)		
1–4, n (%)	70 (35)	14 (43)	14 (30)	36 (33)		
5, n (%)	46 (22)	5 (16)	12 (26)	25 (23)		
≥6, n (%)	89 (43)	13 (41)	20 (44)	48 (44)		
Vomiting, n (%)	118 (58)	22 (69)	28 (61)	57 (28)	* Vomiting/adenovirus	0.788
No. of days with vomiting					norovirus	0.242
1 day, n (%)	64 (54)	7 (32)	16 (57)	36 (63)	rotavirus	0.609
2 days, n (%)	20 (17)	4 (18)	8 (29)	6 (11)	sapovirus	1
≥3 days, n (%)	34 (29)	11 (50)	4 (14)	15 (26)		
Maximum no. vomiting episodes in 24 h						
Median (IQR)	1 (2)	2 (2)	1 (1)	1 (2)		
1 episode, n (%)	61 (52)	7 (32)	16 (57)	30 (53)		
2−4 episodes, n (%)	45 (38)	12 (54)	9 (32)	21 (37)		
≥5 episodes, n (%)	12 (10)	3 (14)	3 (11)	6 (10)		
Fever in previous 48 h, n (%) ^‡^	76 (37)	5 (16)	3 (7)	17 (16)	* Fever/adenovirus	1
≤37.0 °C	171 (83)	26 (81)	43 (93)	88 (81)	norovirus	0.591
37.1–38.4 °C	16 (8)	3 (9.5)	2 (4)	8 (7)	rotavirus	0.099
38.5–38.9 °C	14 (7)	3 (9.5)	1 (2)	9 (8)	sapovirus	0.139
≥39.0 °C	4 (2)	0	0	4 (4)		
Dehydration score						
No dehydration (0), n (%)	20 (10)	3 (9)	5 (11)	10 (9)		
Mild (1–5), n (%)	15 (7)	1 (3)	3 (7)	9 (8)		
Moderate–severe (≥6), n (%)	170 (83)	28 (88)	38 (83)	90 (83)		
Diarrhoea type						
Watery, n (%)	175 (85)	26 (81)	43 (93)	89 (82)		
Dysentery, n (%)	30 (15)	6 (19)	3 (7)	20 (18)		
HIV status					^a^ HIV	
Uninfected, unexposed, n (%)	134 (65)	22 (69)	33 (72)	70 (64)	exposure/	
Uninfected, exposed, n (%)	61 (30)	8 (25)	12 (26)	33 (30)	adenovirus	0.6227 chi^2^ = 0.947
Infected, n (%)	10 (5)	2 (6)	1 (2)	6 (6)	norovirus	0.9062
						chi^2^ = 0.197
					rotavirus	0.6727
						chi^2^ = 0.793
					sapovirus	0.9229
						chi^2^ = 0.160
Baseline Vesikari score, mean (sd)	10.4 (2.7)	11.1 (2.5)	10.1 (2.2)	10.3 (2.9)	^a^ HIV	
Mild (˂7), n (%)	16 (8)	1 (3)	2 (4)	12 (11)	exposure/	
Moderate (7–10), n (%)	88 (43)	11 (34)	24 (52)	45 (41)	Severity of	0.072
Severe (11–20), n (%)	101 (49)	20 (63)	20 (44)	52 (48)	symptoms	chi^2^ = 8.5961

^‡^ Temperature above 38 °C (axillary measurement). IQR-Inter quartile range, sd—standard deviation. * Fisher’s exact test, ^a^ Kruskal-Wallis equality-of-populations rank test.

## Data Availability

Sequence data from this study is available from GenBank, for accession numbers see Section 2.6.

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
