# Peer review of "Virus Etiology, Diversity and Clinical Characteristics in South African Children Hospitalised with Gastroenteritis"

_viruses, 2021, doi:10.3390/v13020215_

Round 1
Reviewer 1 Report
Rossouw and colleagues report the incidence of gastrointestinal viruses in children under 5 years of age from Petroria, South Africa between 2016 and 2017. HIV exposure and FUT2 secretor status were also evaluated. The study is very descriptive and the information collected for secretor status or HIV exposure is not novel. However the limited epidemiological data available for norovirus and astroviruses from Africa warrants the publication of this manuscript. The following comments should be addressed prior to publication.
Comments:
1. The viral sequences reported are very short (particularly those from noroviruses and sapoviruses ~200 bp) and should only be used for genotyping of the viruses detected. This reviewer suggest to omit the phylogenetic analyses (Figures 4-7) as very limited analyses could be done by using those short sequences. Proof of that is the limited description of the phylogenetic analyses made by the authors. Additionally, the discussion of these figures is centered around nucleotide similarity (lines 460-470) and in most part very speculative. Proper analyses require additional sequence information and comparison to all noroviruses sequences available in the GenBank database.
2. Authors reported that "Two GII.4 variants have circulated throughout the study, namely the Sydney 2012 variant, with two different RdRps, GII.P31 (previously GII.Pe) and GII.P16 and an unassigned GII.4, which is related to the Sydney 2012 strain." Please provide additional information about the unassigned GII.4 variant reported. The untyped variant is not shown in Figure 4, and this could be an artifact of the short sequences used in this study. Please confirm by acquiring additional sequence information or omit this from the manuscript.
3. Supplementary Information: Please explain why the sequences for noroviruses are ~200 bp if the expected amplicons are >1 kbp. A better description of the genomic region sequenced should be provided to validate genotyping analyses. For consistency, please provide the expected amplicon for the all primer-pairs used.
4. Lines 168-270: Please update the information of the GenBank submissions.
5. Figure 1b seems unnecessary. The most relevant data is described in the Table 1 and lines 236-239.
6. Line 473: Please correct "was observed at a ratio of 91%:%9 for secretors and non-secretors" to "was observed at a ratio of 10:1 for secretors and non-secretors." Please correct ratio calculations throughout the manuscript.
7. The discussion section could be shorten by avoiding repetition of the results (e.g. 368-373; 387-390; 396-399; 406-409).
Author Response
Reviewer 1
Comments:
- The viral sequences reported are very short (particularly those from noroviruses and sapoviruses ~200 bp) and should only be used for genotyping of the viruses detected. This reviewer suggest to omit the phylogenetic analyses (Figures 4-7) as very limited analyses could be done by using those short sequences. Proof of that is the limited description of the phylogenetic analyses made by the authors. Additionally, the discussion of these figures is centered around nucleotide similarity (lines 460-470) and in most part very speculative. Proper analyses require additional sequence information and comparison to all noroviruses sequences available in the GenBank database.
Response: The authors agree that short sequences are not ideal/useful for in depth phylogenetic analyses. However, the analyses of the norovirus partial capsid and RdRp genes (Figure 4) were primarily performed to indicate the level of diversity within the South African strains and the figures provide a visual overview of the relationship between the various genotypes and secretor status, which is of value. In addition, the analyses indicate the different Sydney 2012 lineages that are associated with the [P31] and [P16] RdRp types. Figure 5 is important to show the relationship between the noroviruses detected in symptomatic and asymptomatic infections. Figures 6 and 7 represent the phylogenetic analyses that were done to genotype the rotaviruses and sapoviruses. Therefore, the authors would prefer to keep these figures, to confirm that significant bootstrap support was obtained for the different clusters representing different virus genotypes. The legends to these figures have been modified to indicate clearly that the phylogenetic analysis was for the purpose of genotyping. In addition, references to figures 4-7 were included in the section 3.3 Virus genotyping
The following section was added to the results to expand the description of Figure 4.
Page 9, lines 303-307
“The partial GII.4 capsid sequences formed two clusters correlating with their respective RdRp types [P31] or [P16]. The majority of GII.4[P31] strains (11/15) were very closely related. Two GenBank strains that were detected in 2016 in Australia and China represent a large number of strains with 99-100% identity over the analysed region identified by BLAST analysis.”
- Authors reported that "Two GII.4 variants have circulated throughout the study, namely the Sydney 2012 variant, with two different RdRps, GII.P31 (previously GII.Pe) and GII.P16 and an unassigned GII.4, which is related to the Sydney 2012 strain." Please provide additional information about the unassigned GII.4 variant reported. The untyped variant is not shown in Figure 4, and this could be an artifact of the short sequences used in this study. Please confirm by acquiring additional sequence information or omit this from the manuscript.
Response: The authors agree with the reviewer that the short sequence is not sufficient to reliably classify an unassigned GII.4 variant. This will be omitted from the manuscript and only the GII.P31/GII.4 Sydney 2012 and GII.P16/GII.4 Sydney 2012 included.
New statement in line 456-457:
“The Sydney 2012 variant, with two different RdRps, GII.P31 (previously GII.Pe) and GII.P16 have circulated during the study [77].”
- Supplementary Information: Please explain why the sequences for noroviruses are ~200 bp if the expected amplicons are >1 kbp. A better description of the genomic region sequenced should be provided to validate genotyping analyses. For consistency, please provide the expected amplicon for the all primer-pairs used.
Response: We agree that this section is not clearly explained. The size of the amplicons differed depending on which gene region was successfully amplified. We obtained longer sequences for some strains, but in order to include all of the sequences in the phylogenetic analysis, the sequences were trimmed to ~ 200 bp for phylogenetic analysis.
To clarify the norovirus genotyping strategy, the following section was added to the Methods on pages 3 and 4 under 2.5.1 Calicivirus genotyping, lines 143-147:
As a first step a ~1100 bp region spanning the 3’-end of the RdRp and the 5’-end of the major capsid genes (AC region) was targeted for amplification. If this region did not amplify, a smaller region spanning the ~560 bp BC region, was amplified. In some cases, the BC region also failed to amplify after which region C (~320 bp) was amplified.
The amplicon size and primer binding sites have been updated in Supplementary Table 1.
- Lines 168-270: Please update the information of the GenBank submissions.
Response: The GenBank accession numbers were provided on page 4, lines 175-177.
“rotavirus VP4 (MW392000-MW392041), rotavirus VP7 (MW392042-MW392081), sapovirus (MW391992-MW391999) and FUT2 (MW491814-MW491833).”
- Figure 1b seems unnecessary. The most relevant data is described in the Table 1 and lines 236-239.
Response: We agree with the reviewer that some of the data in Figure 1b are described in Table 1 and in the text, however, the figure provides information on the distribution of the different viruses in the various age groups which is not apparent from the table or text. For example, the figure shows that co-infections were only detected in children up to 18 months of age, with most co-infections in the 0-6 months age group. Therefore, we would prefer to keep Figure 1b.
- Line 473: Please correct "was observed at a ratio of 91%:%9 for secretors and non-secretors" to "was observed at a ratio of 10:1 for secretors and non-secretors." Please correct ratio calculations throughout the manuscript.
Response: The correction was made as requested on page 16, line 489 and also in three other instances, page 5, line 218; page 13, line 374; page 17, line 529.
The discussion section could be shorten by avoiding repetition of the results (e.g. 368-373; 387-390; 396-399; 406-409).
Response: We have removed the following sections to shorten the discussion as suggested:
Removed original lines 368 – 371:
Overall, 49% of children (n=205) in the study exhibited severe symptoms, with 43% having moderate gastroenteritis and 8% mild cases. Fifty-one percent of children infected with a gastroenteritis virus(es) had severe symptoms, with 45% with moderate symptoms and 4% mild cases.
The revised paragraph (lines 390-395) now reads as follows:
In-depth understanding of the epidemiology of gastroenteritis viruses is crucial to help with the fight against severe viral infections. This 18-month study focused on severe gastroenteritis at a single hospital KPTH, in a relatively urban setting in Pretoria, Gauteng, South Africa’s most populous province. Of the 205 children admitted in the study, the majority (119/205) were under the age of 18 months, underscoring that gastroenteritis is more severe in younger children [56, 57].
Removed original lines 396-397:
Norovirus was the second most prevalent gastroenteritis virus in this study (32/205; 15.6%).
Revised paragraph (lines 415-417):
In lower income countries such as SA, a higher prevalence is more common [61, 62], with a recent study indicating rotavirus prevalence in hospitalised children at 11% [7]. Interestingly severe gastroenteritis was observed in 66% of norovirus cases (21/32) whereas 62.5% (25/40) of the single rotavirus infections were in the mild to moderate category.
Removed original lines 406-409:
Follow up stool specimens were received for 22% (46/205) of the KPTH study population, all of which were collected from children showing no symptoms of gastroenteritis. Norovirus GI (2/10) and GII (8/10) was detected in 22% (10/46) of these specimens.
Revised paragraph (lines 426-430):
Data on asymptomatic norovirus infection is limited in SA. The current study’s results are comparable to a previous study in SA, which detected 36% asymptomatic norovirus infections [64], as well as other studies in low-income settings, such as the Garcia et al. study that found almost 30% norovirus positive specimens from asymptomatic patients in Mexico [65].
Reviewer 2 Report
My comments are available as attachment file.
Comments and suggestions to authors
The authors performed a hospital-based study to assess the distribution and diversity of enteric viruses among hospitalized ≤ 5 years children due to acute gastroenteritis. Authors have also tried to investigate the effect of HIV exposure and secretor status on viral gastroenteritis in South African children. The study findings are informative and of paramount significance for the scientific community as well as the local health department to design and implement appropriate intervention strategies against viral gastroenteritis. However, there are some issues that the authors need to address for the betterment of their research work.
Title:
- The title of the manuscript seems very focused on the effect of HIV exposure and Secretor status on viral gastroenteritis. However, the aim of the study was to determine the distribution and diversity of enteric viruses. Therefore, the title of the manuscript shall capitalize on the main objective of the study rather than the factors which were not well addressed in this study due to insufficient sampling in each cohort (secretor vs non secretor, HI vs HEU vs HU) to assess the effect of the forementioned factors on viral gastroenteritis.
Abstract:
- Authors start the abstract with the study aim. Abstract needs to include some background information in one or two sentences before the study aim to give some background for the readers.
- As the authors presented the abstract section in structured manner, the structure shall include Background, Methods, Results, and Conclusions. In this manuscript, background/introduction is missing as a subtitle under the abstract section which authors shall consider including.
- The sentence “HIV exposure and FUT2 secretor status were evaluated.” presented in the abstract section (line 15-16) shall be moved to the methods section as it is related to the methods.
Methods:
- Authors have explained that cDNA was synthesized for genotyping using ProtoScript II Reverse Transcriptase. However, cDNA was done for RNA viruses only and it should be specified. Therefore, authors shall specifically mention to which enteric viruses cDNA synthesis was done.
- To have appropriate discriminatory between the different genotypes/strains it is recommended to consider around 50% gene coverage for genotyping. However, in this study the targeted gene fragments size (RdRp and capsid genes) for genotyping of norovirus and astrovirus seem short (less than 250 BP). So, with this short fragment size how well the gene/s were represented?
- The authors need to include the gel images of the amplicons used for genotyping of rotavirus and caliciviruses.
- Authors need to address why genotyping was not done for adenoviruses and astroviruses and include it as a limitation.
- Though saliva was collected in this study, the purpose of saliva sample and data from the saliva sample is not included in this manuscript. It would be more informative if the FUT2 genotyping was supported by the HBGA typing in saliva.
Results:
- There is inconsistent presentation of socio-demographic and clinical data in the result section. For example, authors sometimes include the inferential data such as associations of sociodemographic, clinical, and environmental variables (for instance water source) with viral gastroenteritis. While for some of the variables only descriptive statistics (frequency) is considered. I suggest authors to present descriptive statistics (frequencies of all variables) followed by inferential statistics (measures of association) for significant variables in a separate paragraph. Table 1 should also show the p-values and the statistical test values.
- Virus detection rates does not seem correct for sapovirus (n=9, 9%), norovirus GI and astrovirus (n=3, 3% each). Authors need to revise the indicated percentages.
Discussion
- In the first paragraph of the discussion, authors have tried to emphasize that viral gastroenteritis is a major problem despite improvements in water supply, rotavirus vaccine, oral rehydration therapy and reductions in hospitalization and mortality. However, the role of oral rehydration therapy is not clear in reducing the prevalence of viral gastroenteritis rather than improving the clinical outcomes of cases. Therefore, authors need to revise the paragraph in a way that makes more sense to the readers.

Author Response
Reviewer 2
Comments and suggestions to authors
The authors performed a hospital-based study to assess the distribution and diversity of enteric viruses among hospitalized ≤ 5 years children due to acute gastroenteritis. Authors have also tried to investigate the effect of HIV exposure and secretor status on viral gastroenteritis in South African children. The study findings are informative and of paramount significance for the scientific community as well as the local health department to design and implement appropriate intervention strategies against viral gastroenteritis. However, there are some issues that the authors need to address for the betterment of their research work.
Title:
- The title of the manuscript seems very focused on the effect of HIV exposure and Secretor status on viral gastroenteritis. However, the aim of the study was to determine the distribution and diversity of enteric viruses. Therefore, the title of the manuscript shall capitalize on the main objective of the study rather than the factors which were not well addressed in this study due to insufficient sampling in each cohort (secretor vs non secretor, HI vs HEU vs HU) to assess the effect of the forementioned factors on viral gastroenteritis.
Response: The authors have changed the title to the following:
Virus etiology, diversity and clinical characteristics in South African children hospitalised with gastroenteritis.
Abstract:
- Authors start the abstract with the study aim. Abstract needs to include some background information in one or two sentences before the study aim to give some background for the readers.
Response: We have included the following sentence at the beginning of the abstract (Line 13) to include some background:
“Viral gastroenteritis remains a major cause of hospitalisation in young children.”
- As the authors presented the abstract section in structured manner, the structure shall include Background, Methods, Results, and Conclusions. In this manuscript, background/introduction is missing as a subtitle under the abstract section which authors shall consider including.
Response: We have included the subtitle Background at the start of the abstract (Line 13).
- The sentence “HIV exposure and FUT2 secretor status were evaluated.” presented in the abstract section (line 15-16) shall be moved to the methods section as it is related to the methods.
Response: This sentence was moved to the Methods subsection (Line 17-18) of the abstract as suggested.
Methods:
- Authors have explained that cDNA was synthesized for genotyping using ProtoScript II Reverse Transcriptase. However, cDNA was done for RNA viruses only and it should be specified. Therefore, authors shall specifically mention to which enteric viruses cDNA synthesis was done.
Response: We have included the following in the methods to clarify for which viruses cDNA was synthesised (Line 137).
2.5. Genotyping of enteric viruses
2.5.1. Complementary DNA synthesis
Complementary DNA (cDNA) was prepared for norovirus, sapovirus and rotavirus with ProtoScript II Reverse Transcriptase (New England Biolabs, Ipswitch, MA) and random hexamers (Roche Diagnostics Corp., Mannheim, Germany) according to the manufacturer’s instructions using 10 uL of RNA.
- To have appropriate discriminatory between the different genotypes/strains it is recommended to consider around 50% gene coverage for genotyping. However, in this study the targeted gene fragments size (RdRp and capsid genes) for genotyping of norovirus and astrovirus seem short (less than 250 BP). So, with this short fragment size how well the gene/s were represented?
Response: The standard regions targeted for norovirus genotyping include regions A (partial RdRp ~ 300 bp), region BC (~560 bp region spanning the ORF1/2 junction with 3’ end of RdRp and 5’ end of capsid, ORF1/2 overlap) and region C (partial capsid ~320 bp). The final sequences obtained in the study were somewhat shorter than the targeted regions due to low sequence quality at either end of the region. However, all strains could be assigned to a genotype by the online norovirus genotyping tool, and the assigned types were confirmed by our own phylogenetic analysis where the strains clustered with their assigned types with between 80 and 100% bootstrap support. Therefore, we are confident in the assignment of genotypes. Of course, larger genome regions or even complete genomes are ideal, but within this project it was not feasible to determine larger genome regions.
- The authors need to include the gel images of the amplicons used for genotyping of rotavirus and caliciviruses.
Response: We included gel images of the amplicons generated for the caliciviruses and rotavirus genotyping in the Supplementary Information section as Figures S2, S3 and S4.
- Authors need to address why genotyping was not done for adenoviruses and astroviruses and include it as a limitation.
Response: We have included the following statement in lines 521-522 of the discussion that addresses the limitations.
“Funding and time constraints precluded genotyping of adeno- and astroviruses.”
- Though saliva was collected in this study, the purpose of saliva sample and data from the saliva sample is not included in this manuscript. It would be more informative if the FUT2 genotyping was supported by the HBGA typing in saliva.
Response: Saliva was collected with the aim of phenotyping the HBGAs in the children’s samples. Unfortunately, saliva samples of varying quality were obtained from these young children. In many cases saliva was coloured pink. In these samples very high background levels were observed with different anti-HBGA antibodies, therefore the phenotyping results are not considered reliable enough to include in the paper. To avoid confusion in the readers, we have removed the saliva collection from the study description (line 108), since no data is presented on it.
Results:
- There is inconsistent presentation of socio-demographic and clinical data in the result section. For example, authors sometimes include the inferential data such as associations of sociodemographic, clinical, and environmental variables (for instance water source) with viral gastroenteritis. While for some of the variables only descriptive statistics (frequency) is considered. I suggest authors to present descriptive statistics (frequencies of all variables) followed by inferential statistics (measures of association) for significant variables in a separate paragraph. Table 1 should also show the p-values and the statistical test values.
Response: We have rearranged the presentation of the data as suggested by the reviewer. See the two revised paragraphs below:
Lines 213-235
“A total of 221 children ≤ 5 years, hospitalised with gastroenteritis from July 2016 to December 2017 were enrolled. Stool specimens could be collected for 205 children and they were included in the study. The median age of the study population was 10 months (13 days minimum, 60 months maximum), with 82% of specimens (169/205) collected from children ≤ 2 years of age, and the largest contribution from children between 7 and 12 months. The male:female ratio was 1.5:1. The demographic and clinical characteristics of the cohort are summarised in Table 1. Most of the children were HIV unexposed (n=134; 65%), with HIV exposed children accounting for 30% (n=61) of the study population and HIV positive children for 5% (n=10). FUT2 genotyping determined 81% secretors vs 19% non-secretors in this population. The children comprised of 57 homozygous secretors, 110 heterozygous secretors and 38 homozygous non-secretors. The majority of children (71%) exhibited diarrhoeal symptoms for between 1 to 4 days, with 43% of children experiencing > 6 loose/water stools in 24 h. Fifty-eight percent of children vomited and 37% had a fever in the 48 h prior to hospitalisation.
HIV exposure did not affect the frequency of gastroenteritis virus infections in this cohort (p > 0.5). A general trend in increasing ratios of co-infections was observed from HU to HEU to HI. Due to the small HI sample size, this was not statistically significant. The severity of disease was also evaluated against HIV status. The difference in severity of disease for HI, HEU and HU patients was not statistically significant (p=0.072). There was no association between viral gastroenteritis and type of water source (indoor tap compared to outdoor sources, p=0.1993) or type of sanitation (flush toilet versus other types) (p=0.0987).”
P-values and statistical values have been added to Table 1.
- Virus detection rates does not seem correct for sapovirus (n=9, 9%), norovirus GI and astrovirus (n=3, 3% each). Authors need to revise the indicated percentages.
Response: We have corrected these numbers (Line 252):
“sapovirus (n=9, 4%), norovirus GI and astrovirus (n=3, 1.5% each)”
Discussion
- In the first paragraph of the discussion, authors have tried to emphasize that viral gastroenteritis is a major problem despite improvements in water supply, rotavirus vaccine, oral rehydration therapy and reductions in hospitalization and mortality. However, the role of oral rehydration therapy is not clear in reducing the prevalence of viral gastroenteritis rather than improving the clinical outcomes of cases. Therefore, authors need to revise the paragraph in a way that makes more sense to the readers.
Response: We have removed the reference to oral rehydration therapy to avoid confusion. The paragraph now reads as follows (Lines 382-389):
Viral gastroenteritis is still a major cause of illness, in the face of improvements to provision of safe water and sanitation, the rotavirus vaccine and reductions in hospitalisations and mortality [51-53]. In high-income countries, the rotavirus vaccine has led to a rapid decrease in hospitalisations due to rotavirus, with norovirus now the major cause of viral gastroenteritis hospitalisations in children under the age of five [4, 54]. However, in low- and middle-income countries such as South Africa, despite routine rotavirus vaccination, rotavirus is still the leading cause of viral gastroenteritis hospitalisations [55], as was illustrated by this study.